# M3C: A MULTI-DOMAIN MULTI-OBJECTIVE, MIXED-MODALITY FRAMEWORK FOR COST-EFFECTIVE, INDUSTRY SCALE RECOMMENDATION

## ABSTRACT

The ever-expanding landscape of products, surfaces, policies, and regulations poses significant challenges for recommendation systems, leading to data fragmentation and prohibitive hikes in infrastructure costs. To address these challenges, we propose M3C, a holistic co-design of model, data and efficiency strategies. M3C (1) partitions the recommendation space to allow better representation learning and encourage knowledge sharing within a subspace; (2) covers each partition using a hierarchy of foundational and vertical networks tailored to handle multi-domain, multi-objective tasks with mixed-modal inputs; (3) forms a unified data representation that utilizes heterogeneous signals across domains, objectives and optimization goals to alleviate data fragmentation, label sparsity, and to enhance knowledge sharing; (4) improves execution efficiency and lowers costs with a suite of stability and throughput optimizations. We show that across a diverse set of tasks on public and industry datasets, M3C delivers up to 1% lower LogLoss compared to 10 state-of-the-art baselines, while improving system efficiency by up to 20%. Furthermore, in a large-scale industry setting our deployment of M3C has resulted in 7% top-line metrics improvement in online tests with 10% capacity savings.

## 1 INTRODUCTION

High quality recommendation plays a vital role in creating a better online experience. To date, most research in recommendation focuses on improving quality of a single domain-objective pair (Zhang et al., 2024a; Luo et al., 2024; Wang et al., 2021a; Naumov et al., 2019), with the assumption that on-boarding better models for a pair in the pipeline translates to better overall metrics.

However, modern recommendation systems are highly complex, with thousands of domains and tasks, and a typical user request can trigger hundreds of models in the pipeline. This makes the existing upscaling approach cumbersome: the constant innovations in new products and services require recommendation systems to quickly adapt to diverse and heterogeneous user behaviors and to cover new domains and objectives, but given tight inference latency requirements, it becomes unmanageable to introduce a new model for each domain-objective pair, because each model needs to be separately trained, optimized, and served. Further, the changing expectation of users on the usage of their data, the increasing variety of demands from advertisers, as well as rapidly-evolving regulations and policies from the government (Voigt & Von dem Bussche, 2017) and industry unavoidably limit both the amount, quality, and granularity of data available for models, resulting in fragmentation, sparsity and ultimately quality loss for existing frameworks.

To ensure sustainable growth, we must break away from the traditional approach of per domain-objective pair scaling and redesign our strategy around consolidating model, data sources to facilitate knowledge sharing and lower infrastructure cost. However, several challenges stand in the way.

Unlike content understanding, where multi-modal foundational models (Ngiam et al., 2011; Bommasani et al., 2021) can learn inherent representations from diverse data sources using the next token prediction task, data in recommendation is inherently sparse, fragmented, dynamic, and incoherent, making self-supervision unsuitable. Furthermore, the mix of non-sequential (traditional sparse features) and sequential data (user history behavior (Chen et al., 2019)) complicates the development of

Figure 1: M3C consolidates fragmented data, partitions recommendation surface and construct MDMO M3C Networks from which efficient user-facing vertical models are distilled and optimized.

a unified model architecture for all recommendation tasks. Even if an all-in-one model were feasible, its computational costs would likely be prohibitive.

We propose M3C, a novel framework that addresses the challenges in recommendation scalability through a holistic co-design of model, data, and training system. M3C consists of the following:

**M3C Partitioner**: partitions the recommendation surface based on domain, task, optimization goal, and policy regulations, enabling knowledge sharing and better model and objective representations.

**M3C Networks**: a hierarchy of multi-domain, multi-objective (MDMO) models with a novel architecture that handles mixed-modal inputs and reduces costs via knowledge distillation.

**M3C Zipper** and **Filter**: combine heterogeneous data sources to form a coherent feature set, balance label freshness and cost, and select the optimal set of features given a M3C Network.

**M3C Sketch**: an automated tool that leverages scaling laws to optimize hyperparameters and parallelization strategies, improving model latency and throughput without compromising quality.

We evaluate M3C through extensive experiments on real-world recommendation scenarios, using both public and industry-scale datasets. Our results show that M3C significantly outperforms 10 state-of-the-art baselines in terms of model quality and hardware efficiency. Furthermore, our deployment of M3C on a representative set of Ads model types has yielded a 7% increase of top-line metrics in online A/B tests with and a 10% capacity saving.

## 2 RELATED WORK AND CHALLENGES

This section provides a review on the recent advancements and unsolved challenges for industry-scale recommendation in the context of MDMO learning, data strategy and cost efficiency.

### 2.1 MDMO LEARNING

**Status Quo** Current state-of-the-art recommendation systems focus on optimizing a narrow set of objectives (e.g., AUC and LogLoss of CTR) within specific domains (Zhang et al., 2024a; Naumov et al., 2019; Wang et al., 2021a; Cheng et al., 2020; Mao et al., 2023). This traditional approach becomes inefficient as the number of domains and objectives grows, and scaling up isolated models overlooks opportunities for cross-domain knowledge sharing.

Building on top of multi-domain recommendation (Li & Tuzhilin, 2020; Yan et al., 2019; Ma et al., 2018a; Sheng et al., 2021; Wang et al., 2022a; Yang et al., 2022; Tang et al., 2020) and multi-task recommendation (Liu et al., 2022; Malhotra et al., 2022; Li et al., 2023; Yang et al., 2023; Wang et al., 2021b; 2022b), progresses are made on the front of MDMO to mitigate this problem. In particular, M2M (Zhang et al., 2022b) introduces a meta unit, an attention module and a tower module to incorporate domain task knowledge, explicit inter-domain/task correlations and specialize on the task-specific features. $M^3oE$ (Zhang et al., 2024b) learns common, cross-domain information, domain-specific and task-specific information through a mixture of experts, then it leverages a two-level mechanism to aid feature extraction and fusion across tasks. PEPNet (Chang et al., 2023) uses embedding and personalized network to fuse features with different importance and to personalize DNN parameters to balance targets with different sparsity through a novel gating mechanism that incorporates a per-user prior. M3REC (Lan et al., 2023) proposes a meta-learning solution that uses a

meta-item-embedding generator and user preference transformer to unify embedding representation and a task-specific aggregate gate for MDMO learning.

**Challenges** However, prior work does not address the following concerns: (1) scalability: its efficacy on massive, industry-scale recommendation remains uncertain, as most evaluations are limited to a set of domains and objectives; (2) practicality: using a single model to cover a large domain-objective pair space is problematic, as competing domains and objectives can lead to subpar performance (He et al., 2022) or even loss divergence (Tang et al., 2023b); and (3) deployability: consolidating multiple domains and objectives into a single MDMO model can compromise serving latency.

## 2.2 DATA STRATEGY

**Status Quo** Most research on MDMO learning assumes a consolidated dataset available for model training (Zhang et al., 2022b; Chang et al., 2023), this is usually done by partitioning a dataset and designating a few features as prediction tasks to simulate an MDMO (Zhang et al., 2024b; Lan et al., 2023) setting. However, real-world datasets from different domains are fragmented, misaligned, incoherent, sparse, and mixed-modal.

**Challenges** The construction of datasets entails more complex operations than joins: (1) data from multiple products and services, may not have overlapping ID spaces; (2) signals can be gathered across different attribution windows, making it challenging to consolidate the data into a single, unified pipeline without incurring significant costs or compromising on stability and freshness; and (3) the absence of a common self-supervised task and the presense of both sequence and non-sequence inputs adds to the difficulty of a unified representation[1].

## 2.3 COST EFFICIENCY

**Status Quo** Recommendation systems have stringent serving budget and freshness requirements, which entails highly-efficient training and serving. However, recommendation models are notoriously hard to optimize due to large embedding tables (Lian et al., 2021) and many small irregular-shaped kernels, resulting in poor efficiency on modern hardware whose scaling prioritizes compute over network and memory bandwidth (Luo et al., 2018)

Existing work tackles the efficiency problem from multiple fronts. From the angle of creating efficient, scalable architectures, Wukong (Zhang et al., 2024a) adopts compressed dot products and efficient linear compression schemes as its core interaction mechanism; AutoInt (Song et al., 2019) employs the heavily optimized transformer architecture; Mamba4Rec (Liu et al., 2024) uses state-space (Hamilton, 1994) to model historical user behaviors with linear time complexity. Orthogonally, adapting model architecture to datacenter topology proved useful. DMT (Luo et al., 2024) improves distributed embedding lookup performance by adopting multi-rail AlltoAll communication and tower-compressed embeddings to support extra-large embedding tables; NeuroShard (Zha et al., 2023) optimizes for better load-balanced sharding of embedding tables across accelerators to achieve better performance; DHEN (Zhang et al., 2022a) proposes a hybrid sharding strategy to leverage fast NVLink connection in a single host for efficient parameter synchronization. Conversely, adaptation of hardware to recommendation workloads is no longer uncommon (Tal et al., 2024).

On the other hand, training instability due to natural distribution shifts, data corruption (noisy data), and multi-modal learning across a diverse set of tasks also affects cost efficiency. To that end, methods including gradient clipping (Pascanu et al., 2013; Tang et al., 2023b; Wei et al., 2023; Brock et al., 2021; Seetharaman et al., 2020; Menon et al., 2019; Zhang et al., 2019), better feature interaction (Adnan et al., 2023), and more effective normalization techniques (Ba et al., 2016; Santurkar et al., 2018) are proposed.

**Challenges** Despite these advances, improving model efficiency remains a challenge due to the time-consuming optimization-validation cycle even with MDMO learning, as each model still requires independent tuning. Therefore, there is a strong need for automated tooling that can efficiently reason about quality and efficiency tradeoffs.

---

[1]Although orthogonal approaches in generative recommendation via user history modeling (e.g., (Zhai et al., 2024; Sun et al., 2019)) are helping to mitigate this challenge, widespread industrial adoption of this paradigm is likely to take time.

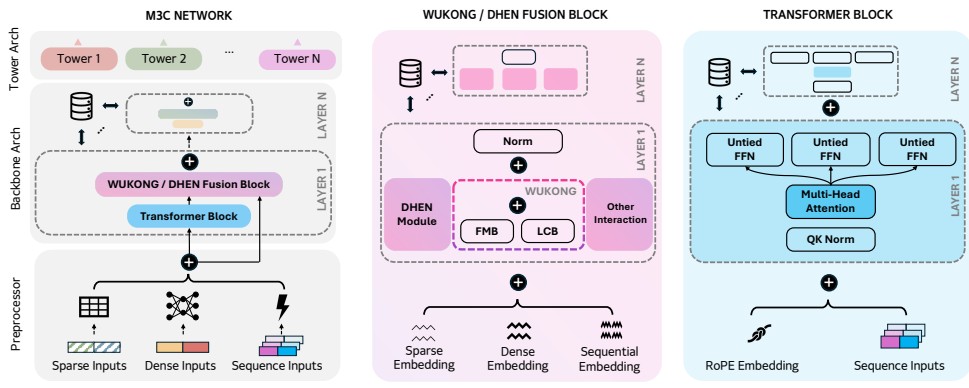

Figure 2: Model architecture of M3C Networks.

# 3 M3C

This section details how M3C systematically tackles the challenges outlined in §2 through a comprehensive co-design of model strategy, data foundation, and cost efficiency optimizations.

## 3.1 MODEL STRATEGY

M3C approaches MDMO by first organizing the domain-objective space into a manageable number of partitions with similar optimization goals to improve representation learning of underlying patterns, then by constructing a hierarchy of a foundational and a vertical *M3C Networks* for each partition.

### 3.1.1 IMPROVING REPRESENTATION VIA M3C PARTITIONER

Squashing all domains and objective into a single model space with poorly designed objective function leads to training instability (Tang et al., 2023a) and subpar performance (He et al., 2022).

**Model Space Representation** To create a better model space representation for MDMO training, M3C Partitioner divides user engagement data into partitions by following these steps:

- First, it partitions domain-objective pairs by the surfaces (e.g., products) they belong. Different surfaces can have distinct data characteristics, which can result in different ID spaces that do not overlap significantly. Such subspaces are less likely to benefit from consolidation, as there is limited knowledge to share between them. For example, it may not be useful to group surfaces that target different age groups together, as they can differ significantly in terms of audience and item pool.
- Next, it groups domains and tasks based on their optimization goals. This considers two key factors: signal staleness and signal density. Signal staleness refers to the time-sensitive nature of certain tasks, such as modeling users' real-time preferences (e.g., click, like, follow). Signal density, on the other hand, refers to the frequency of different events. For example, liking, clicking, and following happen more often than purchasing.
- Finally, policy requirements are taken into account to ensure compliance.

M3C Partitioner then allocates the global budget (e.g., FLOPs and storage) for each partition based on estimated revenue growth potential or expected cost reduction.

**Auxiliary Loss Representation** When domains are merged together, we can merge similar objectives across domains. To ensure the new label mixture correlate well with the original objective, we incorporate an auxiliary loss during training: $Loss(X, Y) = 1 - \frac{Cov(X,Y)}{\sigma_X \sigma_Y}$, where $X, Y$ are the original labels and prediction labels respectively, $Cov$ is the covariance and $\sigma$ is the standard deviation. To ensure the primary task is properly trained with the added auxiliary tasks (Ma et al., 2018b), M3C needs to carefully balance its losses because some tasks can produce gradients of different magnitudes. To address this, we adopt MetaBalance (He et al., 2022) to regulates the gradient scale between the primary and auxiliary tasks.

### 3.1.2 MDMO Learning via M3C Networks

M3C Networks are MDMO models that cover one domain-objective partition. They take in three modalities as inputs: traditional categorical inputs ($\mathcal{F}_c$), dense inputs ($\mathcal{F}_d$), and sequence inputs ($\mathcal{F}_s$, e.g., user history of past interactions) in a batch $B$ then output one prediction for each task. To unify M3C Network's model architecture and to boost MDMO learning across a wide range of input types, M3C Networks adopt a preprocessor-base-tower architecture (Figure 2) to first find common representations for inputs, then perform cross domain interaction, followed by task specialization.

**Feature Processors** serve two purposes: they convert the inputs into dense representations for the backbone, and they unify the representation and project the embeddings into a compatible space so the backbone can efficiently mix and interact across modality. Precisely:

- $\mathcal{F}_c$ are processed through embedding tables, with the output $O_c$ in the shape of $(B, |\mathcal{F}_c|, d)$.
- $\mathcal{F}_d$ are processed by a dense network, which outputs embeddings $O_d$ of shape $(B, |\mathcal{F}_d|, d)$.
- Sequence inputs are created by lightweight event models, which collects sequential items recorded by different surfaces, align/reorders the events, and derives embeddings for the sequence. The event model outputs a sequence embedding $O_s$ of shape $(B, |\mathcal{F}_s| \times K, d)$, where $K$ is the number of recent events to retrieve from each event source (e.g., Ads clicks, post views).
- Optionally, organic sequence data such as texts and image patches are either used as a dense embedding produced by pretrained encoders, or as discretized token-based inputs (Team, 2024).

Now, $O_c$, $O_d$, and $O_s$ share the same format and they are fed into a mixing network, which concatenates $O_c$ and $O_d$ to form $O_{cd}$ as a unified nonsequence data and applies nonlinearities and normalization to it, while leaving sequence data $O_s$ intact, because sequence/non-sequence data are best processed by different modules in the backbone network.

**Backbone** M3C Network Backbones are built with efficient dense scaling (Zhang et al., 2024a; Shin et al., 2023; Zhang et al., 2023) in mind to better tap into modern hardware with superior compute than memory and network capabilities (Luo et al., 2018; 2020). To that end, M3C Networks utilize three components: a supporting structure called extended context storage (ECS), a transformer (Vaswani et al., 2017) block, and a DHEN (Zhang et al., 2022a)/Wukong (Zhang et al., 2024a) fusion block (DWFB). These blocks enable interleaved learning approach for $O_{cd}$ and $O_s$, which proves highly effective for learning combined sequence and nonsequence data.

The ECS is a key-value store accessible globally to simplify implementation of advanced residual connections (Huang et al., 2017) and to allow use of intermediate results in later layers. ECS also provides a means to track statistics across a wide range of metrics for debugging purposes.

The transformer block has the standard architecture proposed by (Vaswani et al., 2017) for sequence learning. Each transformer block corresponds to a stack of transformer layers, with RoPE embedded (Su et al., 2024) input sequence $O_s$, and a contextualized sequence $O_s'$ as output. The current setup of transformer blocks can be viewed as a form of early fusion (Team, 2024), where different user history types are merged into a single event sequence at the event model. However, recent evidence suggests that untying modality can be beneficial (Lin et al., 2024; Zhou et al., 2024). Inspired by these, FFN layer in M3C Network's transformer block can be conditioned on the input to better capture the inherently different underlying patterns. For example, click event sequence and post view sequence do not need to share the same FFN weights. To mitigate training instability issue arising from competing modalities, we apply QK-norm (Henry et al., 2020) before attention.

DWFB is introduced to address deficiency of transformers in processing non-sequence data due to the lack of bit-wise interaction (Wang et al., 2021a; Zhang et al., 2024a) for $O_{cd}$. DWFB flattens $O_s'$, which can be viewed as user-side features, then it concatenates them with $O_{cd}$ as an input and produces a new version of $O_{cd}'$ as the output. DWFB captures feature interactions in $O_{cd}$ in a hierarchical manner. Horizontally, each of the fusion block adopts an intra-layer interaction ensemble of Factorization Machine Block (FMB), Linear Compression Blocks (LCB), and MLPs used in (Zhang et al., 2024a), ensuring each DWFB layer captures both bit-wise and feature-wise interactions. Vertically, DWFB stacks $L$ interaction layers, and each DWFB captures up to $2^{L-1}$ degrees of interaction (Zhang et al., 2024a).

**Tower Modules** Task-specific adaptation in M3C Networks is achieved through tower modules, one for each prediction objective. Tower modules are usually lightweight MLP layers that project the common embeddings learned from the backbone into the task space.

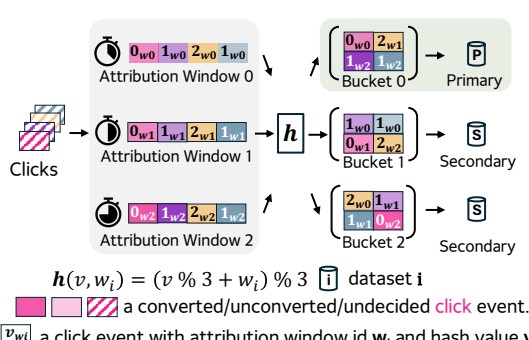

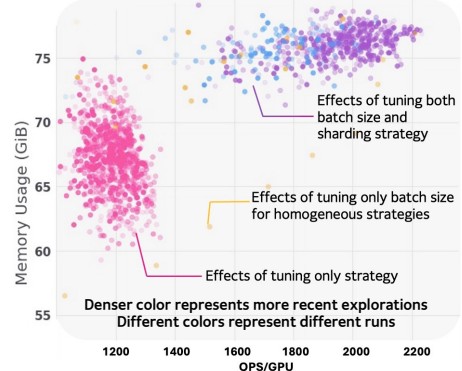

Figure 3: Illustration of M3C Zipper with $K = 3$ for 4 CVR events across 3 attribution windows.

Figure 4: M3C Sketch improves model throughput by 20% on 128 A100 GPUs.

## 3.2 DATA FOUNDATION

Training MDMO M3C Networks requires a coherent feature set to be built from fragmented data sources. This section introduces methods to create such datasets and select optimal features.

### 3.2.1 CONSOLIDATING DATASOURCES VIA M3C ZIPPER

With similar domains and objectives grouped by M3C Partitioner, we still need to cater to customer's specific needs in terms of different attribution windows. Thus, there presents a unique challenge in balancing between data freshness and label completeness: when attribution window is smaller, the data is fresher, but labels may be incomplete; in contrast, when a window is large, more labels are available but they can be stale. An example of this is a CVR event: the advertiser may only let us know whether the customer made a purchase after the attribution window.

However, with $K$ attribution windows ($w_i, i \in [1, K]$), each action $e$ generates $K$ pairs of events, each corresponding to a label of $l_i \in \{0, 1\}, i \in [1, K]$ within a window, we cannot simply include all attribution windows in a dataset because it inflates training cost by $K\times$. Since most events only have a binary label, including multiple pairs for the same event brings no new information while resulting in significant overfitting (Zhang et al., 2022c) or training stability issues (Zhou et al., 2018), because $(e, 0)$ or $(e, 1)$ will appear multiple times across windows. Therefore, we require that (1) each event must appear exactly once in the dataset, with the label associated with one of its attribution windows; (2) the training set size should not increase; and (3) the dataset construction is efficient.

We give $(e, l_i)$ a $\frac{1}{K}$ chance of being selected, with a sequence of $(e, l_i)$ in the order of attribution window $w_1...w_K$, then the probability of $(e, w_i)$ being chosen is $\prod_{j=1}^{i}(1 - \frac{1}{K})^{j-1}\frac{1}{K}$. Unfortunately, implementing this naively requires us to sequentially process each $(e, l_i)$ which is too slow.

We propose M3C Zipper, an optimized algorithm to create such a dataset in parallel. M3C Zipper first assigns each $(e, l_i)$ to the $i$-th worker, then it extracts a set of features $\mathbb{F}_e$ that uniquely identifies $e$. Next, M3C Zipper introduces the same hash function $\mathbf{h}(\mathbb{F}_e) \to \mathbb{Z}$ across workers, which computes the bucket ID that the pair $(e, l_i)$ belongs to as $(\mathbf{h}(\mathbb{F}_e) \bmod K + i) \bmod K$.

M3C Zipper then picks bucket 0 as the dataset. To see how M3C Zipper implementation satisfies the three requirements, first notice that $\mathbf{h}(\mathbb{F}_e)$ is a constant w.r.t $e$, and with any $i, j \in [1, K]$ we have $i \neq k \to i \bmod K \neq j \bmod K$, thus no two attribution of the same event will be assigned to the same bucket; since each event appears $K$ times across $K$ windows and there are same number of buckets as attribution windows, each bucket has exactly $|e|$ items; further, each worker operates independently on each $(e, w_i)$, we can parallel the construction without any synchronization.

It's worth noting that $\mathbb{F}_e$ includes a timestamp, which is a good source for randomness, thus M3C Zipper introduces no bias towards an attribution window for an event. Unlike previous methods for multi-attribution consolidation that primarily focused on continuous training with negative samples (Ktena et al., 2019; Wang et al., 2020) or label correction with reweighing (Chen et al., 2022), M3C Zipper does not require multi-pass training, further boosting efficiency.

### 3.2.2 PARETO-OPTIMAL FEATURE SELECTION VIA M3C FILTER

The dataset constructed by M3C Zipper can contain tens of thousands of features.

However, not all features are required in a subspace created by M3C Partitioner. Given a cost budget and a M3C Network, M3C picks Pareto-optimal features (Mishan, 1967) via M3C Filter. Pareto optimality is a condition in MDMO where no further improvement on one objective can be made without hurting the other objectives. To illustrate how M3C Filter works, we first define feature importance score vector for the i-th feature (in the feature set $\mathcal{F}$) for $N$ tasks as $F_i = (f_{i,1}, \ldots, f_{i,N})$, for $i \in [1, |\mathcal{F}|]$. Each of the $f_{i,j}$ represents the importance score in $j-$th task ($j \in [1, N]$) computed with the permutation importance algorithm (Breiman, 2001). We then define a partial order relationship among all $F_i$s, called *dominated by* $(\mathcal{D}, \preccurlyeq)$ as follows: $F_i \preccurlyeq F_k \iff \forall j \in [1, N], f_{i,j} \leq f_{k,j}$.

Now, given a feature set $\mathcal{F}$ and target feature count $T$ (a few thousands), M3C Filter iteratively finds undominated features on the current pareto frontier of the remaining feature set. In each iteration, up to $T$ of such features are selected and removed from $\mathcal{F}$. The algorithm returns when $\mathcal{F}$ is empty or the target feature count $T$ is reached. Appendix A.1 provides the outline of the algorithm.

M3C Filter can work with heterogeneous tasks thanks to its "unitless" property. Additional feature selection criteria such as cost, coverage, and freshness can be incorporated easily, supporting the joint optimization over MDMO learning and providing rich, compact, and generalizable feature sets.

Previous schemes that directly compute feature importance scores (e.g., using Shapley-Value (Roth, 1988)) then picking top-K features that result lowest loss value (Ma et al., 2018b; Xi et al., 2021; Yasuda et al., 2022) are no longer optimal. Since the loss term is now a combination of losses from different domains and objectives, simply using the global loss value as the surrogate for feature selection unavoidably misses out the opportunity to leverage knowledge sharing across tasks and leads to over-exploiting tasks that are easier to optimize.

### 3.3 EFFICIENCY OPTIMIZATIONS

This section details how M3C improves cost-efficiency of M3C Networks.

### 3.3.1 STABILIZING M3C NETWORK TRAINING

Instability issues in MDMO training can arise from various aspects. For example, on mixed-modal datasets contention among different modalities cause each modality to increase its norm (Team, 2024). To address this, we apply aggressive normalization techniques including QK-norm and layernorm in the mixing network when the inputs are from different interaction modules. Additionally, we apply adaptive gradient clipping (Tang et al., 2023a) for dense parameters to further stabilize training.

### 3.3.2 IMPROVING EXECUTION EFFICIENCY VIA M3C SKETCH

M3C Networks are trained via hybrid parallelism (Mudigere et al., 2021) on a GPU cluster, where the embedding tables are sharded across different devices via TorchRec (Ivchenko et al., 2022), and dense parameters are synchronized by Fully Sharded Data Parallel (FSDP) (Zhao et al., 2023).

To further improve model execution efficiency without affecting quality, we propose a search framework, called M3C Sketch, that unifies the search for optimal model hyperparameter and parallelization strategies given cost budgets without hurting model accuracy.

**Search Space** The M3C Sketch search space is defined by a model template with undecided hyperparameters, parallelization strategies, and their associated choices or ranges.

**Objectives** Quality-related signals are slow to obtain. To accelerate the search process, M3C Sketch leverages established scaling laws (Shin et al., 2023) of Wukong and Transformers to approximate model quality. In particular, given a FLOPs budget $f$, Wukong's logloss improves linearly to $f^{0.00071}$ (Zhang et al., 2024a), and Transformer's logloss scales linearly to $f^{-0.05}$ (Kaplan et al., 2020). In Wukong, we simultaneously scale output embedding count in its LCB, FMB and dot interaction compression factor to hold the law; in transformer, Narang et al. (2021) has shown that basic modifications to the architecture do not result in significant quality changes. Thus, M3C Sketch

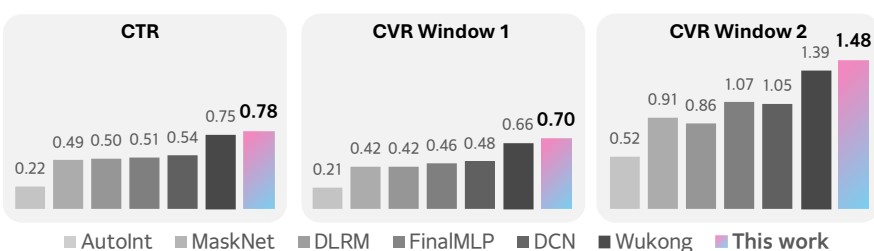

Figure 5: Relative LogLoss Improvement (%) over AFN on an industry-scale dataset.

formulates the objective by incorporating throughput and quality proxy using scaling laws to find optimal configurations. Appendix A.2.1 provides details.

**Constraints** To simplify search, M3C Sketch encodes constraints in the objectives. For unsatisfied constraints, a reward of negative infinity is used. M3C Sketch constraints can include FLOPs input and hyperparameter dependency.

**Search Strategy** M3C Sketch uses an alternating solving strategy to approximate the optimal solution while significantly reducing search time via beam search of width $K$. M3C Sketch first fixes parallelization strategies by randomly samples $S$ configurations while searching for the best model hyperparameters. The top $K$ found best model hyperparameters are then selected for the next round, which become the fixed parameters while parallelism strategy is mutated. This process repeats until either search quota is depleted or the results no longer improve.

M3C Sketch uses parallel Bayesian optimizers (Snoek et al., 2012) that periodically merges the search trajectories during the search steps. To further guide the search process, M3C Sketch adopts the performance model used in Srifty Luo et al. (2022) and uses a dynamic programming algorithm (Appendix A.2.2) for bootstrapping.

## 4 EVALUATION

We demonstrate the effectiveness of M3C then ablate contribution of each component. We include detailed setups for each experiment in the Appendix section to improve reproducibility.

### 4.1 EVALUATION SETUP

We evaluate M3C on one public and two internal datasets and compare with 10 state-of-the-art baselines including AFN+ (Cheng et al., 2020), AutoInt+ (Song et al., 2019), DLRM (Naumov et al., 2019), DCNv2 (Wang et al., 2021a), FinalMLP (Mao et al., 2023), MaskNet (Wang et al., 2021c), xDeepFM (Lian et al., 2018), BST (Chen et al., 2019), APG (Yan et al., 2022) and Wukong (Zhang et al., 2024a). The public dataset is an representative competition dataset released by Kuaishou (Kuaishou) as used in (Zhang et al., 2024a; Li et al., 2019; Zhu et al., 2023). The two internal datasets are with and without sequence data, with 3K and 1K features respectively to assess multi-modality performance. We report AUC and LogLoss for public dataset and LogLoss improvement for internal datasets over a baseline as quality metrics. To ensure fair comparison, we turn off distillation for all models. See Appendix A.3.1 for detailed experimental setup.

### 4.2 MODEL PERFORMANCE GAINS

**Open Source Dataset: KuaiVideo** We evaluate baselines and M3C Networks on the KuaiVideo dataset for MDMO performance, predicting the labels for $like$, $follow$, and $click$. We report final test performance in Table 1. We highlight best and second-performing models using bolds and underlines. Our expert-tuned M3C Network matches or outperforms baseline performance in AUC and LogLoss with comparable complexity. See Appendix A.4.1 for model details.

**Industry-Scale Dataset: Uni-modality** We scale up selected models to 30-40 GFlops (about $1000\times$ compared to those used in the KuaiVideo) and evaluate them on our industry-scale dataset with 100B data. We focus on three tasks: one CTR task and two CVR tasks across two attribution windows.

| Model | AUC | | | | LogLoss | | | | Complexity | |
|---|---|---|---|---|---|---|---|---|---|---|
| | Click | Follow | Like | AVG | Click | Follow | Like | AVG | MFLOPs | MParams |
| AFN+ | 0.7172 | 0.7703 | 0.8466 | 0.7780 | 0.4572 | 0.0074 | 0.8466 | 0.1604 | 10.96 | 79.60 |
| AutoInt+ | 0.7181 | 0.7882 | 0.8725 | 0.7929 | 0.4607 | 0.0075 | 0.0156 | 0.1617 | 79.27 | 41.75 |
| DLRM | 0.7088 | 0.6743 | 0.7734 | 0.7188 | 0.4874 | 0.0081 | 0.0175 | 0.1710 | 1.996 | 39.24 |
| DCNv2 | 0.7225 | 0.7954 | 0.8804 | 0.7995 | 0.4534 | **0.0073** | 0.0152 | 0.1586 | 9.159 | 40.44 |
| FinalMLP | 0.7176 | 0.7627 | 0.8624 | 0.7809 | 0.4690 | 0.0080 | 0.0163 | 0.1645 | 12.22 | 571.4 |
| MaskNet | 0.7133 | 0.7143 | 0.8599 | 0.7625 | 0.4650 | 0.0077 | 0.0156 | 0.1627 | 4.299 | 39.63 |
| xDeepFM | 0.7189 | 0.7704 | 0.8706 | 0.7866 | 0.4642 | 0.0079 | 0.0156 | 0.1626 | 6.810 | 51.60 |
| APG (DeepFM) | 0.7066 | 0.7464 | 0.8515 | 0.7682 | 0.4915 | 0.0080 | 0.0166 | 0.1720 | 11.74 | 52.40 |
| BST | 0.7217 | 0.7664 | 0.8707 | 0.7863 | **0.4512** | 0.0076 | 0.0153 | 0.1581 | 326.1 | 40.63 |
| Wukong | 0.7251 | 0.7947 | 0.8842 | 0.8014 | 0.4580 | 0.0075 | 0.0155 | 0.1603 | 22.62 | 42.59 |
| M3C Network (*) | **0.7281** | **0.7997** | 0.8793 | 0.8024 | **0.4513** | **0.0073** | 0.0154 | **0.1580** | 25.54 | 43.08 |
| M3C Network (+) | 0.7249 | 0.7984 | **0.8861** | **0.8031** | 0.4529 | **0.0073** | **0.0151** | 0.1584 | 1.575 | 39.17 |

Table 1: Test performance on KuaiVideo across 3 tasks. M3C models are tuned to maximize AVG AUC on the validation dataset with 3 tasks. *: tuned by expert; +: autotuned by M3C Sketch.

The result is summarized in Figure 5. Evidently, M3C Networks continue to significantly outperform other state-of-the-arts. Appendix A.4.3 provides details on these models.

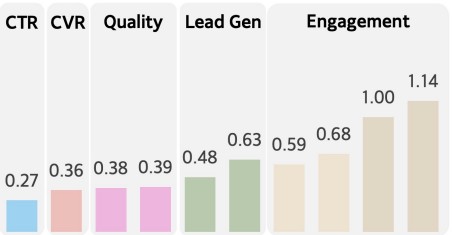

Figure 6: Relative LogLoss Improvement (%) of M3C Network over Wukong on an industry-scale, mixed-modal dataset.

Figure 7: Relative LogLoss Improvement (%) of M3C Zipper across various dates in a month, compared to an ideal upperbound.

**Industry-Scale Dataset: Mixed-modality** With a mixed-modal dataset (50B samples), we focus on comparing M3C Network with scaled-up Wukong. We use the same event models in M3C Network to construct sequence digests, and feed these as user-side features into Wukong so it can reason about sequences. We evaluate both M3C Network and Wukong on 10 tasks spanning CTR, CVR, lead generation, quality, and engagement. We summarize the outcome in Figure 6. While M3C architecture incorporates Wukong as its interaction block, it significantly outperforms Wukong in every category with comparable complexity, signaling the effectiveness of interleaved sequence/nonsequence learning. See Appendix A.4.4 for details on the model specifications.

### 4.3 DATA STRATEGY EFFECTIVENESS

**M3C Zipper** We evaluate M3C Zipper on a CVR-focused model with 90-min/1-day (short/long) attribution window separately, on a Wukong-based model (See Appendix A.4.5 for specs). We estimate the limit of improvement by modeling an ideal case where all labels are available immediately as "Upperbound". For all experiments, we use 1:1[2] sampling ratio for short/long join window pipeline. We evaluate the model on different date ranges under daily incremental recurring training to provide long-term signal. Figure 7 shows the consistent improvements of M3C Zipper across multiple evaluation dates in a month. Since existing approaches for delayed feedback modeling (Wang et al., 2020; Chen et al., 2022) cause severe training diverge or loss regression, we exclude them from comparison.

**M3C Filter** We implement M3C Filter on top of pre-generated per-task feature importance result to get about 2K candidates out of 12K features. We use default weighted loss feature selection logic as the baseline and evaluate the relative LogLoss improvements on the same dataset using 10 models spanning 4 optimization targets. We summarize the metric wins and generalizability of M3C Filter in Table 2. Appendix A.4.6 provides a description for the setups.

---

[2]We observe only minor improvements by further tuning this ratio.

| | CTR | CVR | CTR+CVR Consolidation | CTR+Quality Consolidation |
|---|---|---|---|---|
| **Relative LogLoss Improvement (%)** | $0.2 \sim 0.5$ | $0.12 \sim 0.17$ | $0.1 \sim 0.5$ | $0.06$ |

Table 2: M3C Filter evaluation on CTR/CVR Tasks

## 4.4 COST EFFICIENCY IMPROVEMENTS

**M3C Sketch on KuaiVideo for Quality Improvements** We setup a search space for M3C Sketch of 4 trillion choices and apply it to M3C Network on the KuaiVideo dataset. We set the objective to maximize $O_{VM} = max(\frac{AUC}{f0.003})$ on the validation set. We use 8 parallel M3C Sketch solvers with a budget of 1200 steps. We pick the best configuration and present results in Table 1. The resulting model achieves state-of-the-art results in 4 out of 8 metrics, outperforming the expert tunings with $17\times$ fewer FLOPs. Appendix A.4.2 provides details.

**M3C Sketch for Throughput Improvements** Finally, we evaluate M3C Sketch on an 8-layer M3C Network with the objective to maximize throughput by simultaneously adjusting the batch size and FSDP parallelization strategy for each DWFB on 128 Nvidia A100 GPUs. The search space is around 300B. We summarize the exploration trajectory of M3C Sketch in Figure 4. While batch size has larger implication on the throughput than tuning sharding strategies alone, combining both in the search space leads to the most significant gain: M3C Sketch finds the best hyperparameters that improves throughput by 20% compared to the best expert-tuned baseline, and by 10% when tuning parallelization strategies only. Appendix A.4.7 details the search space and model specifications.

## 5 INDUSTRY-SCALE DEPLOYMENT AND SUSTAINABILITY IMPACT

As recommendation model sizes increasing by $20\times$ in recent years in real world deployments (Wu et al., 2022; Zhai et al., 2024), surging demand has made it the single largest AI application in terms of infrastructure demand in the datacenters of major Internet companies (Mudigere et al., 2021; Park et al., 2018). To reduce energy footprint and improve serving quality, we deployed M3C on a representative set of Ads model types in a large-scale industry setting.

**Model Space and Data Source Consolidation** M3C is applied to a representative set of domain-objective pairs formed with thousands of domains and tens of objectives from our own services and advertiser goals into a very small set of M3C groups via M3C Partitioner, supported by unified data sources constructed by M3C Zipper and M3C Filter. This drastically reduced the amount of models required from hundreds to a small, manageable number, thereby significantly reducing the compute demand to support the entire space without losing quality.

**Reducing Model Footprint via Knowledge Distillation** To meet stringent serving latency requirements, M3C constructs a hierarchy of teacher and lighter-weight, user-facing student M3C Networks to transfer knowledge from teacher to student via label-based distillation(Hinton et al., 2015). Distillation converts the serving *latency* problem into a *bandwidth* problem: the teacher throughput only needs to match the data volume during the refresh interval of a student in online training, which can be satisfied by adjusting the training scale. Label based distillation requires minimal storage overheads, however, M3C Networks use more aggressive distillation via feature-based distillation (Romero et al., 2014; Heo et al., 2019) for critical tasks to improve distillation effectiveness at a higher storage cost.

**Results** M3C has delivered a 7% top-line metric gain in online A/B tests and 10% capacity savings at the same time. We anticipate greater benefits as we continue to roll out M3C to fully unleash its power to accelerate technology incubation, enable faster product growth, offer agility in shifting market landscapes, deliver improved user satisfaction, all with a greener approach.

## 6 CONCLUSION

M3C co-designs novel network, data and efficiency strategies to consolidate recommendation surface, model space and data sources to attain state-of-the-art MDMO quality gains as well as cost and resource reductions. In particular, the evaluation on public KuaiVideo as well as industry uni- and mixed-modality datasets has shown that M3C delivers up to 1% lower LogLoss while improving system efficiency by up to 20%. Furthermore, our deployment of M3C in a large-scale industrial environment has resulted in improvement of top-line metrics by 7% with 10% capacity savings.

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

# A  APPENDIX

## A.1  THE M3C FILTER ALGORITHM

Algorithm 1 summarizes the M3C Filter algorithm.

---
**Algorithm 1** The M3C Filter Algorithm
---
1: **function** FEATURE_FILTER($T, \mathcal{D}$)
2:     counter ← Dict()
3:     dominates ← Dict()
4:     **for** $f_i, f_j$ in $\mathcal{D}$ **do**
5:         counter[$f_i$] ← counter[$f_i$] + 1
6:         dominates[$f_j$] ← dominates[$f_j$] $\cup \{f_i\}$
7:     **end for**
8:     ss = SortedSet((d_count, f) for (f, d_count) in counter.items())
9:     selected = []
10:     **while** $T > 0$ **do**
11:         _, f = ss.pop(0)
12:         selected ← selected $\cup \{f\}$
13:         **for** $f\_dominated$ in dominates[$f$] **do**
14:             ss ← ss $\setminus \{(\text{counter}[f_{dominated}], f_{dominated})\}$
15:             counter[$f_{dominated}$] ← counter[$f_{dominated}$] - 1
16:             ss ← ss $\cup \{(\text{counter}[f_{dominated}], f_{dominated})\}$
17:         **end for**
18:         T ← T - 1
19:     **end while**
20:     **return** selected
21: **end function**

---

## A.2  M3C SKETCH

### A.2.1  M3C SKETCH OBJECTIVES

For complex M3C Networks with a large FLOPs budget we can use compute complexity as a proxy for model quality. However, since FLOPs budget is provided as a total sum between Transformer and Wukong blocks, expert insight is required to determine the allocation ratio. For less complex models, we continue to incorporate a quality term directly in the objectives.

Specifically, for complex models, M3C Sketch uses $O_{M3CNetwork} = max(\frac{f}{t})$ as the objective, where $f$ is the FLOPs complexity, $t$ is the latency. Given a FLOPs target $F$ and a maximum iteration latency $T$ derived by dividing the minimum throughput required for online training, we have the constraints of M3C Network as $\alpha F \leq f \leq \frac{1}{\alpha}F$ (model quality control) and $t \leq T$ (minimum throughput limit). We use $\alpha \in [0, 1]$ as a threshold hyperparameter to M3C Sketch which gives the search more wiggle room. For less complex models, the objective becomes $O_{VM} = max(\frac{q}{F^\beta})$, with $q$ being the model quality evaluated after fixed amount of training (e.g., 1B data) is done, and $\beta$ being the quality scaling ratio, and any hyperparameters resulting quality to scale superlinearly with respect to FLOPs is more rewarded.

### A.2.2  BOOTSTRAPING M3C SKETCH

We follow the same maximum batch size estimation strategy as used in Srifty Luo et al. (2022). For bootstrapping the search process of FSDP configurations, given a collection of supported sharding strategies $\mathcal{X}$ for each FSDP unit, and for a model with $L$ layers training on $W$ GPUs with $K$ valid batch sizes, and $\mathcal{D}_W$ as the set of factors of $W$. We use dynamic programming to statically find a reasonable starting point for the search: given a model, we first profile the execution latency $T_b(s_l)$ and memory usage $R_b(s_l)$ of $l$-th layer given a batch size $b$ and a FSDP sharding strategy $s_l \in S = \mathcal{X} \times \mathcal{D}_W$, which can be obtained in parallel across GPUs. We then estimate batch-

independent communication latency for $l$-th layer using data volume and network bandwidth $C(s_l)$. Finally, we solve this problem using dynamic programming.

Let $R$ be the GPU memory capacity, $W$ be the number of GPUs in the fleet, $ANS_b[i, r]$ be the best sharding strategies we find for when only considering layers up to $i$, with the maximum allowed memory consumption of $r$, and $OPT_b[i, r]$ be the corresponding execution latency, then we have the best configuration $ANS_b^*$ derived by:

1. Border condition:
$$OPT_b[l, r] = \infty, ANS_b[l, r] = 0, \forall_x x \leq 0 \vee r > R \vee x > L$$

2. Recurrence:
$$s_{l+1,r}^* = argmin_{s \in S} OPT_b[l, r - R_b(s)] + T_b(s) + C(s)$$
$$OPT_b[l + 1, r] = min_{s \in S} OPT_b[l, r - R_b(s)] + T_b(s) + C(s)$$
$$ANS_b[l + 1, r] = ANS_b[l, r - R_b(s_{l+1,r}^*)] \leftarrow s_{l+1,r}^*$$

3. Final output:
$$ANS_b^* = ANS_b[L - 1, argmin_r OPT_b[L - 1, r]]$$

We can simply enumerate $b$ to find the global minimum execution latency and its corresponding FSDP configuration, and the overall complexity of this algorithm is bound by $O(|K| \times LR|S|)$. Since the performance models are analytical, this seed is not necessarily optimal in practice Luo et al. (2022), hence the M3C Sketch process continues, guided by the Bayesian optimizer.

### A.3 DETAILED EXPERIMENTAL SETUP

#### A.3.1 DATASETS

We evaluate M3C on both public and internal datasets. The public dataset KuaiVideo is an representative competition dataset released by Kuaishou Kuaishou as used in Zhang et al. (2024a); Li et al. (2019); Zhu et al. (2023). This dataset has 13M entries with 8 features. We use this dataset as a standard benchmark for recent state-of-the-art models in multi-objective recommendation. We use the train/test split provided by the BARS Zhu et al. (2022) benchmark suite, and we further perform 9 to 1 train and validation split. We use and extend the FuxiCTR framework Zhu et al. (2021) for experimentation on public dataset. We use two internal datasets, with and without sequence features to evaluate model performance. The dataset without event feature has about 1K features. The dataset with event features contains roughly 2K features and 9 event sources. For evaluating data strategy, we use an internal dataset with about 2K nonsequence features, selected from a pool of 12K features.

#### A.3.2 METRICS AND OBJECTIVES

For KuaiVideo, we report AUC and logloss; for internal datasets, we report improved normalized entropy (NE) He et al. (2014) over a baseline, following prior arts.

Due to lack of a common fundamental task across all recommendation tasks, we use a collection of representative tasks to *approximate* a foundational task from which all downstream tasks can benefit. For the Kuaishou dataset, we predict three tasks: *is_like*, *is_follow* and *is_click* and use the average loss as the final loss. For internal dataset, we create 3 tasks derived from a combination of click and conversion rate prediction across various attribution windows, and we train each model sufficiently long enough to draw conclusions and report metrics.

#### A.3.3 BASELINES

We focus on comparing with recent state-of-the-art recommendation models including AFN+ Cheng et al. (2020), AutoInt+ Song et al. (2019), DLRM Naumov et al. (2019), DCNv2 Wang et al. (2021a), FinalMLP Mao et al. (2023), MaskNet Wang et al. (2021c), xDeepFM Lian et al. (2018), BST Chen et al. (2019), APG Yan et al. (2022) and Wukong Zhang et al. (2024a). For public dataset, we use the best-tuned config from BARS if available, otherwise we use the configuration used by Zhang et al. (2024a). For internal dataset, we use the model tunings adopted by Zhang et al. (2024a). We

| Template | Choices |
|---|---|
| Layers | 1,2,4,8,16,2x2,2x3,2x5,2x7,3x1,3x2,3x3,4x2,4x3 |
| Each Activation Function | RELU, GELU, SILU, TANH |
| Each Dropout | 0, 0.01, 0.05, 0.1 |
| Batch Norm | True, False |
| Each Projection Dimension | 128, 256, 512, 1024, 64, 32, 16, 8, 4 |
| Each Projection Layers | 1,2,3 |
| k | 8, 16, 32, 64, 4, 2, 1 |
| Embedding Regularizer | 0.0001,0.0002,0.00005,0,0.00001,0.0005,0.001,0.1,0.5,1 |
| Total | 4.1 Trillion Choices |

Table 3: M3C Sketch template and associated search space.

| Model | Hyperparameters | Valid AUC |
|---|---|---|
| M3C (Expert Tuned) | l=2 (3 + 1 Wukong), nL=6, nF=1, k=16, MLP=256 | 0.8020 |
| M3C (M3C Sketch) | l=1, nL=3, nF=4, k=4, MLP=512 | 0.8045 |

Table 4: Best model configuration for KuaiVideo found by experts and M3C Sketch.

| Model | Hyperparameters | GFLOPs | Parameters (B) |
|---|---|---|---|
| AFN+ | DNN=4x8192, afn=4x8192, nlog=4096 | 43.40 | 633.95 |
| AutoInt+ | Attention=3x512, nhead=8, DNN=3x8192 | 42.53 | 631.49 |
| DCN | l=2, rank=512, MLP=4x32768 | 43.88 | 634.46 |
| DLRM | TopMLP=4x16384 | 31.07 | 632.39 |
| FinalMLP | MLP1=4x16384, MLP2=2x4096, output_dim=64 | 36.26 | 633.27 |
| MaskNet | MLP=3x2048, nblock=4, dim=128, reduction=0.05 | 32.36 | 632.67 |
| Wukong | l=8, nL=32, nF=32, k=24, MLP=3x16384 | 35.16 | 633.08 |
| M3C | l=12, nL=96, nF=96, k=96, MLP=8192-4096-8192 | 32.16 | 632.46 |

Table 5: Detailed hyperparameters, compute complexity and model size for each run used in the evaluation of different models on a uni-modal internal dataset.

report model complexity and parameter count for fair comparison. Note that we may use different model configurations to highlight the effectiveness of different components, which we detail in their respective sections.

## A.4 MODEL SPECIFICATIONS

### A.4.1 MODEL SPECIFICATION FOR KUAIVIDEO DATASETS

For all baselines, we use the hyperparameters tuned by FuxiCTR and Wukong. We use the same batch size of 64K and the same learning rate for all models. We turn on FuxiCTR's learning rate schedule and set patience to 5 for tuning on the validation set.

### A.4.2 M3C SKETCH ON KUAIVIDEO FOR QUALITY IMPROVEMENTS

See Table 4 for details on the best found model by M3C Sketch on the KuaiVideo dataset and the search space employed by M3C Sketch.

### A.4.3 MODEL SPECIFICATIONS FOR A UNI-MODAL INTERNAL DATASETS

See Table 5 for details on the models used in internal dataset experiments.

| Model | Hyperparameters | GFLOPs | Parameters (B) |
|---|---|---|---|
| Wukong | l=5, nL=128, nF=128, k=64, MLP=6144-3072-6144 | 26.93 | ≈ 934.14 |
| M3C | l=2, Wukong=2/layer, nL=128, nF=128, k=64, MLP=6144-3072-6144 Transformer=1/layer, MHA.head=2, FFN=256 | 23.43 | ≈ 934.44 |

Table 6: Detailed hyperparameters, compute complexity and parameter count for models used in the evaluation on a mixed-modal internal dataset.

| Model | Hyperparameters | GFLOPs | Parameters (B) | Features |
|---|---|---|---|---|
| Wukong | l=2, nL=32, nF=32, k=32, MLP=2048-512-2048-1024-2048 | 0.726 | 998.60 | ≈ 1800 |

Table 7: Model specification for M3C Zipper experiments.

### A.4.4 MODEL SPECIFICATIONS FOR MIX-MODAL INTERNAL DATASETS

See Table 6 for specifications of the M3C and Wukong models used in the experiment.

### A.4.5 MODEL SPECIFICATIONS FOR M3C ZIPPER EXPERIMENTS

See Table 7 for details on the models used in internal dataset experiments.

### A.4.6 MODEL SUMMARY FOR M3C FILTER EXPERIMENTS

We evaluated M3C Filter on 10 models spanning 4 optimization targets (4 models for CTR, 4 models for CVR and 1 model each for CTR+CVR and CTR+Quality consolidation) to obtain the generalizability and effectiveness of M3C Filter across a wide range of models and tasks. Each model is configured similarly to those used in Appendix A.4.5.

### A.4.7 M3C SKETCH ON WUKONG FOR THROUGHPUT IMPROVEMENT

See Table 9 for details on the best configuration by M3C Sketch and search space employed in the throughput improvement experiments.

| Syntactical Blank | Choices |
|---|---|
| Each FSDP Wrapping | Z3 (ZERO3), Z2 (ZERO2) |
| Each FSDP Wrapping Replication Group | 1, 2, 4, 8, 16 |
| Batch Size | 1024-4096 |
| Total | 307.2 Billion Choices |

Table 8: M3C Sketch Syntactical blanks and associated search space.

| Model | Hyperparameters FSDP Wrapping Strategies | GFLOPs | Dense Parameters B |
|---|---|---|---|
| M3C (Expert Tuning) | l=8, nL=96, nF=96, k=32, MLP=4096-2048-4096-2048-4096 (Fixed) L1-8: Z3@128; Batch: 1024 | | |
| M3C (M3C Sketch) 1 | l=8, nL=96, nF=96, k=32, MLP=4096-2048-4096-2048-4096 (Fixed) L1,3,6,8: Z2@128; L4,5: Z3@128; Batch: 1280 | 34.80 | 3.99 |
| M3C (M3C Sketch) 2 | l=8, nL=96, nF=96, k=32, MLP=4096-2048-4096-2048-4096 (Fixed) L1,2,7,8: Z2@4; L3,6: Z2@128; L4,5: Z3@128; Batch: 1024 | | |

Table 9: Best parallelization strategy for the internal model found by M3C Sketch.

