# OpenReview forum: "M3C: a Multi-Domain Multi-Objective, Mixed-Modality Framework for Cost-Effective, Industry Scale Recommendation"
_ICLR.cc/2025/Conference — ICLR 2025 Conference Withdrawn Submission_

### Official Review · Reviewer_53pG · 2024-10-20

**Soundness:** 3
**Presentation:** 2
**Contribution:** 2
**Rating:** 3
**Confidence:** 4

**Summary:**

This paper introduces M3C, a framework designed for industrial recommender systems utilizing multi-objective optimization and multi-modality data. M3C comprises four core modules aimed at tackling challenges related to data handling and training efficiency. Experiments were conducted on three real-world datasets, where the experimental results demonstrated its superiority over compared baselines.

**Strengths:**

1. The studied problem is important for industry. The intuition is clear and reasonable. The background and challenges are thoroughly illustrated.
2. The core modules of M3C, including Partitioner, Networks, Zipper and Filter, and Sketch, are mostly reasonably
designed to address the challenges mentioned.
3. Empirical experimental results revealed it superior performance over several baselines.

**Weaknesses:**

1. The novelty is somewhat limited. It primarily focuses on addressing specific engineering challenges within a particular domain by integrating multiple existing models and techniques. This work appears more appropriate for submission to the industrial track of a data mining conference such as SIGIR or KDD.
2. The novel aspects of this framework primarily reside in the design of the Zipper, Filter, and Sketch modules. However, the explanations and illustrations regarding these components are sometimes unclear. I found it challenging to grasp certain details of the Zipper algorithm, such as the role of "workers" and the rationale behind extracting the set of features Fe. Figure 3 was also confusing to me. Similarly, understanding the pseudocode in Algorithm 1 was difficult. Furthermore, Section 3.3.2 glosses over many details of the M3R Sketch, while Appendix A2.2 remains difficult to comprehend and follow. The authors don’t disclose plans to release the code and data, making it more difficult to understand the details and reproduce the model.
3. A recent published work, “M3oE: Multi-Domain Multi-Task Mixture-of Experts Recommendation Framework”, shares quite similar theme and technical approach. While it is acknowledged in the related works, further detailed discussion and comparison appear necessary.

**Questions:**

All my questions are listed in the weakness part.

---

### Official Review · Reviewer_XYjV · 2024-10-28

**Soundness:** 3
**Presentation:** 2
**Contribution:** 3
**Rating:** 5
**Confidence:** 3

**Summary:**

This paper proposes a framework for Multi-Domain Multi-Objective Mixed-Modality (M3C) in industrial recommendation systems. The framework primarily addresses the challenges of MDMO learning, data strategy, and cost efficiency by introducing various components, including the M3C Partitioner, M3C Networks, M3C Zipper and Filter, and M3C Sketch. These components are designed to tackle the mentioned problems effectively. Experiments conducted on both public and industrial datasets demonstrate the effectiveness of the M3C framework.

**Strengths:**

S1: This paper focuses on industrial applications, specifically utilizing internal datasets with thousands of features, offering a realistic reference for real-world scenarios.

S2: The experiments are validated on both public and industrial datasets and are compared with ten SOTA baselines, demonstrating the framework's effectiveness.

**Weaknesses:**

W1: The structure and writing of this paper require significant improvement. The absence of a framework overview makes it difficult to understand the overall structure of M3C, hindering the ability to follow each component's function throughout the recommendation system pipeline.

W2: The paper does not adequately address the challenges outlined in Section 2. Issues such as the ‘non-overlapping ID spaces’ and the ‘optimization-validation cycle’ are not explicitly discussed.

W3: In this paper, does the author acknowledge that "MIXED-MODALITY" refers to various types of features, such as sparse, dense, and sequential features, rather than the commonly recognized inputs like images, videos, or audio?

W4: Some important details are either too brief or missing, such as the Knowledge Distillation of M3C Networks and RoPE embedding.

**Questions:**

See Weaknesses.

---

### Official Review · Reviewer_ddv7 · 2024-11-02

**Soundness:** 3
**Presentation:** 2
**Contribution:** 3
**Rating:** 5
**Confidence:** 4

**Summary:**

The paper proposed a recommendation framewokr M3C that co-designs novel network, data and efficiency strategies to consolidate recommendation surface, model space and data sources to attain state-of-the-art MDMO quality gains as well as cost and resource reductions. Experiements showed promising results.

**Strengths:**

1. the paper discussed the heavy load of data consolidation and computation density, the proposed optimizations are reasonable.
2. the experiments are detailed and their results are sound.

**Weaknesses:**

1. The proposed M3C consists of many components but without ablation studies, making it hard to judge the benefits each componenet brings.
2. The choices of design and strategies are heavily depended on user scenarios. Readers without the corresponding background can hardly understand the paper. E.g., I doubt few academia folks have learnt the term "attribution window".

**Questions:**

1. The M3C has included WUKONG/DHEN. Is it necessary? What's the rationale behind it.
2. Is M3C easy to reproduce?

---

### Official Review · Reviewer_q5Wi · 2024-11-03

**Soundness:** 3
**Presentation:** 2
**Contribution:** 2
**Rating:** 3
**Confidence:** 4

**Summary:**

This paper introduces a systematic framework for multi-domain, multi-objective, and multi-modal recommendation. The framework encompasses several key components, including domain-objective pair partitioning, external context storage, a Transformer and DWFB-based model architecture, a zipper and filter mechanism, and Bayesian optimization for hyperparameter tuning. While the proposed approach demonstrates superior performance in terms of Logloss compared to the baseline Wukong model on an internal dataset, there are several areas that require further attention.

**Strengths:**

-	The authors have validated the performance enhancements over WuKong using an industry-scale dataset, which showcases the potential of practical application.

-	The paper provides a clear and thorough discussion and analysis of related works, which aids in understanding the research motivation.

**Weaknesses:**

-	The introduction is well-written and engaging; however, the description of the framework lacks depth in certain areas. For instance, the domain-objective pairs partitioner and the mechanism for grouping these pairs are not clearly articulated. This omission makes it challenging to assess the flexibility of the proposed method and to understand how it balances commonalities and distinctions among pairs. The same problem applies to ECS. It is highly recommended that the authors provide detailed mathematical formulations or pseudo-code for this section.

-	There seems to be an omission regarding the multi-domain setting. It would be helpful to know on which domains the proposed method learns and effectively transfers knowledge.

-	The choice of baselines, except for Wukong, are somewhat outdated. It would be beneficial to include comparisons with more recent state-of-the-art multi-domain and multi-task baselines to position the proposed method.

-	Considering that Wukong is utilized as component of the model, there seems trivial performance gap between the proposed method and Wukong in Table 1, even the proposed method is tuned by M3C Sketch. It deserves further ablation study to evaluate the contribution of each component.

**Questions:**

Please refer to the weaknesses

---

### Note · Authors · 2024-11-20

**Comment:**

We thank the reviewers for the valuable suggestions, and we aim to improve the writing and technical depth of the submission in another venue.

**Withdrawal Confirmation:**

I have read and agree with the venue's withdrawal policy on behalf of myself and my co-authors.